# Effect of *Clostridium butyricum* on Gastrointestinal Infections

**DOI:** 10.3390/biomedicines10020483

**Published:** 2022-02-18

**Authors:** Tadashi Ariyoshi, Mao Hagihara, Motomichi Takahashi, Hiroshige Mikamo

**Affiliations:** 1Department of Clinical Infectious Diseases, Aichi Medical University, Nagakute 480-1195, Aichi, Japan; t.ariyoshi@miyarisan.com (T.A.); hagimao@aichi-med-u.ac.jp (M.H.); motomichi.takahashi@miyarisan.com (M.T.); 2Miyarisan Pharmaceutical Co., Ltd., Saitama City 331-0804, Saitama, Japan; 3Department of Molecular Epidemiology and Biomedical Sciences, Aichi Medical University, Nagakute 480-1195, Aichi, Japan

**Keywords:** gastrointestinal infection, *Clostridium butyricum*, gut dysbiosis, *Clostridioides difficile*, *Helicobacter pylori*

## Abstract

*Clostridium butyricum* is a human commensal bacterium with beneficial effects including butyrate production, spore formation, increasing levels of beneficial bacteria, and inhibition of pathogenic bacteria. Owing to its preventive and ameliorative effects on gastrointestinal infections, *C. butyricum* MIYAIRI 588 (CBM 588) has been used as a probiotic in clinical and veterinary medicine for decades. This review summarizes the effects of *C. butyricum*, including CBM 588, on bacterial gastrointestinal infections. Further, the characteristics of the causative bacteria, examples of clinical and veterinary use, and mechanisms exploited in basic research are presented. *C. butyricum* is widely effective against *Clostoridioides difficile*, the causative pathogen of nosocomial infections; *Helicobacter pylori*, the causative pathogen of gastric cancer; and antibiotic-resistant *Escherichia coli*. Accordingly, its mechanism is gradually being elucidated. As *C. butyricum* is effective against gastrointestinal infections caused by antibiotics-induced dysbiosis, it can inhibit the transmission of antibiotic-resistant genes and maintain homeostasis of the gut microbiome. Altogether, *C. butyricum* is expected to be one of the antimicrobial-resistance (AMR) countermeasures for the One-health approach.

## 1. Introduction

Gastrointestinal infections, caused by the ingestion of pathogens and disruption of normal microbiota [1,2], mainly manifest as clinical syndromes, including acute vomiting, diarrhea, and enteric fever. These infections are induced by viruses, bacteria, protozoa, or parasites, such as norovirus, Shigella spp., *Vibrio cholerae*, *Listeria*
*monocytogenes*, Enterohemorrhagic *Escherichia coli* (EHEC), *Clostridioides difficile*, and *Salmonella* spp. [3,4]. Some of the pathogens contain virulence factors, such as enterotoxins and flagella, which cause increased fluid secretion and decreased fluid absorption in the gut [5].

In previous randomized clinical studies and experimental studies using animal models, probiotics were found to reduce the severity of gastrointestinal infections and modulate host immunity [6,7,8]. Gram-positive *Lactobacillus* spp. and *Bifidobacterium* spp. have been widely used to treat or prevent gastrointestinal infection-associated diarrhea [9]. Additionally, many studies have reported that the short-chain fatty acids (SCFAs) produced by intestinal microbiota in the gut microbiome can affect host immune homeostasis and intestinal barrier function [10,11,12].

Similar to lactate, butyrate is one of the SCFAs produced as end-products of intestinal microbial fermentations [13,14]. Butyrate is rapidly absorbed in the gut and acts as a signaling molecule in receptor-mediated signaling in numerous cell types [15]. Prior to extensive efforts to sequence the gut microbiota, microbial butyrate production in the human gut was already known [16]. However, among the butyrate-producing bacteria, only *Clostridium butyricum* has been used as a probiotic for symptoms associated with gastrointestinal infections.

Despite accumulating evidence regarding *C. butyricum*, reviews of its effectiveness for gastrointestinal infections are limited compared with *Bifidobacterium* spp. and *Lactobacillus* spp. [17,18,19,20,21,22]. Therefore, this review seeks to discuss the excellent clinical efficacy and mechanism of *C. butyricum*, without limiting animal species, especially for gastrointestinal infections, which are frequently reported.

## 2. *Clostridium butyricum*

*C. butyricum*, a historical and beneficial symbiote, butyrate-producing, spore-forming, obligate anaerobe, and Gram-positive rod shape bacterium, is found in a variety of environments, including soil. *C. butyricum* is also detected in the human gut of approximately 20% of adults [23]. *C. butyricum* is symbiotic with its host, resides in the intestinal tract, and grows by fermenting dietary fiber and other materials that are not degraded by the host. During the fermentation process, *C. butyricum* mainly produces butyrate via the *but-buk* pathway [24].

In 1933, Dr. Miyairi isolated *C. butyricum* for the first time from the feces of healthy individuals [25]. Thereafter, *C. butyricum* MIYAIRI 588 (CBM 588), which was isolated from soil in 1963 [26], was formulated and has been widely used in Japan as a drug for gastrointestinal symptoms, such as diarrhea [27]. The safety of CBM 588 has been confirmed not only in preclinical studies under the good laboratory practice (GL) [28,29,30,31] and/or humans but also in broilers and pigs, and it has been used worldwide as a feed additive [32].

In Europe, CBM 588 is mostly prescribed for as a animals feed additive with improvement of zoo technical performance; however, it has been authorized for use as a novel food by the Council and the European Parliament [33].

Seki et al. [27] reported the preventive effect of CBM 588 for antibiotic-associated diarrhea (AAD) in children. CBM 588 was found to protect the gut mucin layer from antibiotic-induced dysbiosis [34,35]. Further, CBM 588 is known to promote mucin production by modulating the gut microbiota [36,37]. As intestinal epithelial cells under the mucin layer contain absorptive epithelial cells [38], a dysfunction of these cells leads to diarrhea induced by incomplete absorption of water in feces.

CBM 588 regulates the expression of various inflammatory and anti-inflammatory cytokines. CBM 588 also induces the differentiation of IL-17-producing γδ T cells, which are known to play a central role in the expression of mucins and tight junction proteins (TJs) in colonic epithelial cells [39,40]. Treatment with CBM 588 attenuates gut inflammation by altering host lipid metabolism due to the upregulation of protectin D1, an anti-inflammatory lipid mediator [41].

CBM 588 modulates the composition of the intestinal microbiota. Previous studies revealed that CBM 588 administration increased beneficial bacterial populations, such as *Lactococcus* spp., *Lactobacillus* spp., and *Bifidobacterium* spp. in the gut microbiota after antibiotic administration [37,40]. These protective effects of CBM 588 are expected to prevent pathogen colonization and enable the treatment of diarrhea in clinical practice.

## 3. Gut Dysbiosis and Gastrointestinal Infections

Gut dysbiosis is the constitutive and functional transformation of the gut microbiota by environment- and host-related factors [42], such as diet, chemotherapy, antibiotic therapy, stress, environment, infection, and genetic factors. The mammalian gut microbiota is a highly abundant and diverse microbial community that resides in the gastrointestinal tract [43]. When microbial diversity is reduced, the competition between bacteria is eliminated and some bacteria may grow abnormally.

For example, high oxygen concentration in the inflamed intestine allows aerobic respiration by Enterobacterales; however, this condition inhibits the growth of Bacteroidia and Clostridia, which are ubiquitous anaerobes. The abnormal growth of Enterobacterales promotes enterobactin production, a potent iron chelator that inhibits the bactericidal action of neutrophil myeloperoxidase. Additionally, local oxidative stress-causing agents and toxins produced by pathogens disrupt tight junctions in intestinal epithelial cells, resulting in the breakdown of the barrier function and the induction of a “leaky gut” [44,45,46].

Dysbiosis-induced mucosal damage leads to a decrease in the regulators of the innate immune system, such as Toll-like receptors (TLRs) and NOD-like receptors (NLRs), enabling pathogen invasion. Thereafter, a reduction in resistance due to the decreased production of mucus, antibacterial peptides, and immunoglobulin A (IgA) antibodies result in colonization by pathogenic bacteria [47,48].

## 4. Bacterial Gastrointestinal Infections

### 4.1. Clostridioides difficile Infection

*C. difficile* is an anaerobic, spore-forming, Gram-positive rod bacteria that mainly uses succinic acid, primary bile acids, and sialic acid as a source of nutrition for growth in the colon [49]. During the growth phase, *C. difficile* produces toxin A/B (TcdA/TcdB) to target Rho proteins in the host cell. Consequently, the actin cytoskeleton, which is normally maintained by RhoA, is disrupted, causing cell death, mucosal injury, and inflammatory cytotoxicity [50,51] (Figure 1).

*C. difficile* infection (CDI) is one of the major causes of nosocomial diarrhea worldwide. CDI associates with gut dysbiosis and causes mild diarrhea to severe colitis [52]. Previous clinical study reported that approximately 90% of pseudomembranous enteritis and 25–33% of antibiotic-associated diarrhea are attributed to CDI [53]. Additionally, CDI increased patient morbidity and decreased quality of life. Consequently, CDI prolonged hospitalization [54,55,56].

In a clinical study, the co-administration of CBM 588 and vancomycin had a beneficial effect on the treatment of CDI [57]. Among 71 patients with CDI, the co-administration of CBM 588 and antibiotics reduced the stool frequency on the second day of treatment (vancomycin alone group vs. vancomycin plus CBM 588: 3.9 times/day vs. 2.6 times/day, *p* < 0.05). Furthermore, compared with vancomycin monotherapy, the co-administration of CBM 588 and vancomycin shortened the treatment periods from 10.9 days to 8.9 days. However, no similar effect was demonstrated by co-administration with *Enterococcus faecium* product.

In in vivo experiments with the CDI model, the incidence of diarrhea was significantly decreased in CBM 588-treated rats [58]. Additionally, CBM 588 reduced the mortality of mice [59] as well as cytotoxin production [59]. In an in vitro study, *C. difficile* decreased cytotoxin production when co-cultured with CBM 588. Further, the proliferation of *C. difficile* was suppressed [60]. Interestingly, these effects showed only at viable CBM 588, which means living and growth cells, contacted to the *C. difficile* cells. Moreover, it is considered that the direct contact with the metabolites of CBM 588 suppressed the toxin production and growth of *C. difficile*. Such findings suggest that CBM 588 attenuated symptoms related to CDI (Figure 1).

Competition for nutrient sources during the bacterial growth phase is one of the mechanisms by which probiotics enhance the resistance to pathogen colonization. Probiotic bacteria and normal gut microbiota are reported to inhibit *C. difficile* growth by consuming nitrogen-containing amino acids, sialic acid, succinic acid, and host-derived glycans, which are nutrient sources for *C. difficile*, and by producing SCFAs [61,62,63] (Figure 1).

Hagihara et al. reported that CBM 588 not only modulated gut microbiota, but negatively modulated gut succinate levels to prevent the growth of *C. difficile* and downregulate tumor necrosis factor-α (TNF-α), ultimately producing macrophages in the colon lumina propria (cLP), which led to a significant decrease in colon epithelial damage. CBM 588 also upregulated T cell-dependent pathogen specific IgA by IL-17A producing CD4^+^ cells and plasma B cells in the cLP. Th17 cells in the cLP were found to promote the gut epithelial barrier function, ultimately enhancing the colonization resistance of *C. difficile* [61] (Figure 1).

The secretion of substances that inhibit the production or activity of TcdA/TcdB counteracts toxicity to enhance *C. difficile* colonization resistance [64,65]. Bacteriocins, a class of antibiotic peptides, are direct weapons that can be secreted by probiotics [66]. As this bacteriocin is only effective against analogous bacteria, it has gained attention for eliminating the target bacteria without disturbing the microbiota (Figure 1).

Recent studies have shown that the stimulation of host immunity by CBM 588 is effective against CDI. By administering CBM 588 to CDI-infected mice, Hayashi et al. demonstrated that (1) butyrate produced by CBM 588 acted as an antibiotic peptide for Reg3βγ, (2) butyrate-induced neutrophils were not only induced through GPR43/109a signaling but also by metabolites besides butyrate, and (3) Th1 and Th17 cells are induced by butyrate-mediated GPR43/109a signaling [67] (Figure 1).

### 4.2. Helicobacter pylori Infection

*Helicobacter pylori* is a Gram-negative, helical, microaerophilic, flagellate bacterium [68]. This pathogen is known as a gastric carcinogen, and its eradication is associated with the incidence of gastric cancer [69,70,71]. However, *H. pylori* has excellent defense mechanisms, such as gastric acid resistance by urease and drug resistance by biofilm formation [72,73] (Figure 2).

Mukai et al. retrospectively evaluated the eradication rate in 468 patients with *H. pylori* infection. Based on their results, the rate was significantly higher in patients administered proton pump inhibitors (PPIs) and CBM 588 (87.1%) than those administered PPIs alone (70.1%) [74]. Additionally, the co-administration of CBM 588 and PPI reduced the incidence of diarrhea or soft stool during *H. pylori* eradication therapy (43% in the control group and 14% in the CBM 588 normal dose group, and 0% in the CBM 588 double dose group) [74].

Although the number and detection rate of *Bifidobacterium* spp. and ectopic anaerobes were found to be reduced by the antibiotic eradication therapy [75], the number of ectopic anaerobes in the CBM 588 double-dose group was significantly higher than that in the control group [75]. Thus, CBM 588 can contribute to the maintenance of gut microbiota homeostasis and improve the bactericidal effect against *H. pylori* (Figure 2).

Chen et al. reported an increase in *Bacteroides* spp. and a decrease in gastrointestinal symptoms with CBM 588 administration [76]. Bacteroides, a member of the S24-7 family, has been reported to increase the number of innate lymphoid cells 2 (ILC2) in the stomach. Further, IL-5 released from ILC2 stimulates B cells to produce IgA [77]. However, the mechanism involved in the improvement of the eradication rate of *H. pylori* owing to the combination of probiotics, including CBM 588, is unclear (Figure 2).

The effect of CBM 588 on *H. pylori* infection has been reported based on basic research. Takahashi et al. [78] reported that CBM 588 culture supernatant inhibited the growth of *H. pylori*; this inhibitory effect was observed even when the culture supernatant was adjusted to pH 7, indicating a dependence on the butyrate produced. Furthermore, the inoculation of *H. pylori*-infected gnotobiotic mice with CBM 588 reduced the number of *H. pylori* bacteria to less than 1/100 [78].

The adhesion of *H. pylori* to gastric epithelial cells is a primary event in the development of infection. As *Lactobacillus* spp. inhibit the adhesion of *H. pylori* to MKN45 and Caco-2 cells [79,80], CBM 588 displayed an inhibitory effect on the adhesion of *H. pylori* to MKN45 cells [81]. However, it is still unclear how CBM 588 inhibited the attachment of *H. pylori* to gastric epithelial cells (Figure 2).

The gut microbiota and its metabolites are markedly altered in patients after gastrectomy [81]. Future studies are expected to investigate the effects on the stomach owing to the changes in the gut microbiota. Most bacteria are useful as probiotics colonize the colon while *H. pylori* colonize the stomach. Hence, it is unlikely that probiotics are directly involved in the lurking of *H. pylori* in the stomach. Nevertheless, the enhanced eradication effect may be attributed to the improvement of systemic immunity due to immunostimulation by probiotics [82,83] (Figure 2).

### 4.3. Escherichia coli Infection

*E. coli* is a facultative anaerobic, Gram-negative rod-shaped bacterium. This bacterium is usually found in the intestines of animals and healthy people. Most types of *E. coli* are harmless and induce mild diarrhea in a relatively shorter period of time. However, *E. coli* O157:H7, can cause severe abdominal pain, vomiting, bloody diarrhea, and stomach cramps [84,85] (Figure 3).

Additionally, enterohemorrhagic *E. coli* (EHEC), one of *E. coli* strain, causes serious intestinal infection. EHEC produces a potent toxin called Shiga toxin. This toxin causes bloody diarrhea after distractions of the lining of the intestinal wall [84,85]. EHEC outbreaks have also been linked to some types of foods, as well as surface water areas that animals visited frequently.

Although no clinical study has evaluated the efficacy of *C. butyricum* to EHEC infection, Fujita et al. [86] reported that butyrate produced by CBM 588 reduced the titer of thermophilic enterotoxin regardless of pH in an in vivo study; however, the mechanisms remain unclear. Additionally, the therapeutic administration of CBM 588 to EHEC infected mice reduced the mortality by 50%. Thereafter, prophylactic administration of CBM 588 reduced the mortality to 0%. CBM 588 also reduced the number of EHEC in the gut and the toxin titer in the feces of infected mice. These results suggest that CBM 588 has preventive and therapeutic effects against EHEC [87] (Figure 3).

In vitro studies revealed that the co-culture of CBM 588 and EHEC inhibited the growth of EHEC and reduced their toxin production [88]. Butyrate was found to be involved in these inhibitory effects, and caused pH-independent and dose-dependent antibacterial effects on EHEC. The preincubation of CBM 588 with Caco-2 cells inhibited the establishment of EHEC [87] (Figure 3).

Kunishima et al. [88] investigated the effects of CBM 588 on the growth, β-lactamase activity, and transmissibility of the antibiotic resistance properties of antimicrobial-resistant (AMR) organisms, including extended-spectrum β-lactamase (ESBL)-producing *E. coli* and carbapenem-resistant Enterobacterales. Consequently, the growth of AMR bacteria was inhibited in a dose-dependent manner by the supernatants of *Clostridium* spp. containing CBM 588 (Figure 3).

The β-lactamase activity produced by *E. coli* was found to be reduced in the presence of CBM 588 culture supernatant. Further, the transcription of the blaCTX-M gene is repressed during the growth phase of *E. coli*. A conjugation assay revealed a decrease in the transmissibility of antibiotic resistance genes by enteric bacteria. These results suggest that CBM 588 can be employed to suppress AMR bacteria [88] (Figure 3).

### 4.4. Staphylococcus aureus Infection

*S. aureus* is a Gram-positive round-shaped bacterium and a member of the body’s microbiota as it is frequently found in the upper respiratory tract and the skin [89]. *S. aureus* isolates can produce a variety of enterotoxins and enterotoxin-like substances [90]. Hence, the presence of *S. aureus* in the gastrointestinal tract can result in colonization, food-borne disease, enterocolitis, and toxic shock syndrome [91].

As a recent report suggested that *C. butyricum* alleviates intestinal injury through epidermal growth factor receptor (EGFR) [92,93], Ma et al. [94] evaluated the inhibitory effects of the recombinant strain of *C. butyricum* overexpressing EGF. The recombinant strain significantly inhibited the growth of co-cultured *S. aureus*. An inhibitor was then used to block STAT3 tyrosine phosphorylation, decreasing the antibacterial effect of the recombinant strains.

Compared with the wild-type strain, the recombinant strain increased the expression levels of intestinal formation-related genes (*Claudin-1*, *GLUT-2*, *SUC*, *GLP2R*, *EGFR*) and anti-inflammatory genes (*IL-10*) in intestinal epithelial cells [94]. Hence, the secretory overexpression of pEGF in *C. butyricum* could upregulate the expression level of EGFR, consequently improving the intestinal protective functions of *C. butyricum* partly following STAT3 signal activation in IPECs, causing a positive loop.

### 4.5. Vibrio cholerae Infection

*V. cholerae* is a facultative anaerobic, Gram-negative bacilliform bacterium [95]. This bacterium is found in marine and brackish waters. Additionally, this bacterium adheres to the chitinous shells of shellfish such as shrimp and crabs. Some strains of *V. cholerae* are pathogenic to humans and cause cholera, a fatal disease that originates from the ingestion of raw or undercooked marine species [96].

Kuroiwa et al. [97] investigated the inhibitory effect of CBM 588 on various enteric pathogens in vitro. CBM 588 inhibited the growth of *V. cholerae* O1, *V. cholerae* non-O1, *Aeromonas hydrophila*, and *Shigella flexneri* in co-culture. Furthermore, the inhibitory effect was observed when the pH was kept neutral, suggesting that not only the low pH conditions, but also the metabolites produced contributed to the inhibitory effect of each pathogen [97].

### 4.6. Salmonella Species Infection

*S. enterica* is an aerobic, Gram-negative flagellated and rod-shaped bacterium. *S. enterica* causes most salmonellosis originated from infected foods [98]. To develop infectious diseases induced by *S. enterica,* secreted proteins are important. *Salmonella* spp. can mediate biofilm formation and readily contact with host cells, because of a very large number of fimbrial and nonfimbrial adhesins. Their secreted proteins also play a role in host cell invasion and intracellular growth.

CBM 588 suppressed enteritis caused by *Salmonella* spp. in farm pigs and significantly reduced mortality [99]. Additionally, in an in vivo study with specific pathogen-free (SPF) broilers infected with *S. enteritidis*, *C. butyricum* altered the gut microbiota composition and increased the α-diversity [100]. Further, *C. butyricum* caused the downregulation of inflammation cytokine levels (IFN-γ, IL-1β, IL-8, TNF-α) in intestinal tissues and upregulation of muc-2 and ZO-1 expression levels [100].

## 5. Conclusions

The cause of some gastrointestinal infections can be explained by microscopic mechanisms in the intestinal tract where dysbiosis occurs. Hence, maintaining gut microbiota homeostasis is one of the main purposes of ameliorating some symptoms of gastrointestinal infections. It is also known that the damage to the intestinal tract can be caused by abnormal growth of pathogenic bacteria and associated toxin production [101,102].

Previous in vivo and in vitro study revealed that CBM 588 would be effective to inhibit the proliferations of bacteria and can cause gastrointestinal infections [57,58,59,60,61,62,63,64,65,66,67,74,75,76,77,78,79,80,81,82,83,86,87,88,92,93,94,97,99,100]. Additionally, CBM 588 is a very safe drug since the bacteria has no toxin-producing genes and has been used clinically for over 70 years [25,26,27,28,29,30,31,32]. Therefore, CBM 588 is expected to be effective not only for diarrhea in dysbiosis caused by antibiotics but also for diarrhea in dysbiosis caused by pathogenic bacteria.

By now, no clinical evidence has shown that CBM 588 can reduce pathogenic bacteria in gut by itself. However, there are some supportive in vivo and in vitro study data showing that CBM 588 would be effective in inhibiting the proliferations of bacteria and can cause gastrointestinal infections by butyrate production, nutritional competition, and production of antibiotic substances such as bacteriocins [61,66,67,78,86,87].

Additionally, we are thinking that one of the therapeutic purposes to use probiotics is to inhibit the recurrence and colonization of pathogenic bacteria. Then, some clinical studies have suggested that *C. butyricum* strains not only inhibit many pathogenic bacteria growth, but also enhance host intestinal immunity.

In the case of clinical study, the co-administration of CBM 588 with antibiotics before the onset of diarrhea significantly reduced the incidence of diarrhea [27]. Furthermore, in EHEC-infected rabbit models, prior administration of *C. butyricum* inhibited the growth of EHEC and reduced the incidence of diarrhea and the lethality [103]. In addition, CBM 588 administration prior to CDI induced Th1, Th17, and neutrophils and enhanced *C. difficile* colonization resistance, thereby reducing the lethality [68]. Hence, prior colonization of *C. butyricum* in gut is expected to inhibit the invasion of pathogenic bacteria and to maintain the homeostasis of the microbiota.

SCFAs produced by gut microbiota are known to contribute to host health by regulating intestinal immune homeostasis [104,105,106]. Among them, butyrate can not only inhibit pathogenic bacteria growth by lowering pH, but also serves as a source of energy for mucosal cells in the colon [107]. Butyrate stimulates receptors in the colon to promote intestinal peristalsis [108,109,110], and promotes mucus secretion in the colon, ultimately contributing to the inhibition and elimination of pathogenic bacteria in the intestinal tract [111].

Butyrate has been reported to activate the host SCFAs receptors, such as GPR43 and 109a, and to exhibit anti-inflammatory effects [112]. Butyrate also directly promotes the differentiation of Tregs by inhibiting histone deacetylases (HDACs) [113,114]. Taken together, butyrate-producing *C. butyricum* is very beneficial to the regulation of gastrointestinal infections and further explorations are needed to identify new targets for the prevention of gastrointestinal infections.

We focused on other fatty acid metabolites related to SCFAs and their receptors. By administering CBM 588 to mice with antibiotic-induced gut dysbiosis, we analyzed the immune response of the host intestinal tract, intestinal microbiota, and fecal metabolites [115]. Consequently, the following results were obtained. 1: CBM 588 increased the production of anti-inflammatory omega-3 fatty acids and anti-inflammatory lipid mediators by modulating gut microbiota; and 2: CBM 588 affected lipid metabolites and increased the conversion of linoleic acid metabolites, which are ligands for the fatty acid receptor, GPR120 [115].

Changes in the lipid metabolism of gut microbiota and the host were confirmed by the administration of CBM 588 [115]. The unsaturated fatty acids induced by CBM 588 are expected to contribute to the termination of inflammatory symptoms in gastrointestinal infections. Therefore, the metabolites in feces and gut microbiota should be analyzed and discussed based on the results of multi-omics research.

Although this review has specifically discussed bacterial gastrointestinal infections, some studies have shown that probiotics such as *Lactobacillus* spp. and *Bifidobacterium* spp. have antiviral effects [116,117,118]. Evidence showing the anti-viral effect of *C. butyricum* has not been confirmed. However, CBM 588 induces cytokines and immune cells with anti-viral effects, including IFN-γ and/or IgA, in various models. Hence, CBM 588 may indirectly inhibit virus infection through Lactobacillus and Bifidobacterium species proliferations [61,67]. Moreover, protectin D1, the metabolite induced by CBM 588 in antibiotic causing dysbiosis model, inhibits viral RNA transport to the nuclear envelope, thereby suppressing influenza virus replication [119].

In conclusion, butyrate-producing CBM 588 is presently making a significant contribution to the endless number of fatalities caused by gastrointestinal infections. Alleviating the crisis caused by gastrointestinal pathogens in animals, including humans, is the mission of future probiotic products. Additionally, *C. butyricum* can serve as one of the AMR countermeasures, enabling us to tackle future global problems, as this bacteria can inhibit the transmission of antibiotic-resistant genes and maintain gut microbiota homeostasis.

## Figures and Tables

**Figure 1 biomedicines-10-00483-f001:**
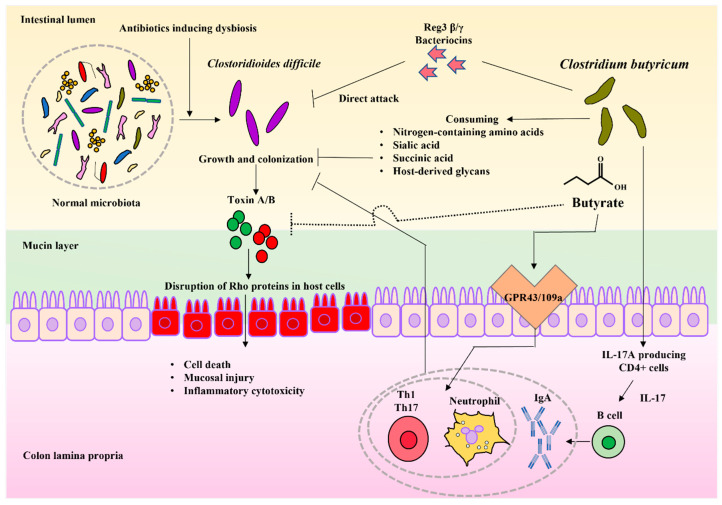
The mechanisms whereby *C. butyricum* protects against *Clostridioides difficile* infection: *C. difficile* survives when treated with antibiotics and becomes predominant during dysbiosis. *C. difficile infection* (CDI) is established by colonization and toxin A/B production, which disrupts the cytoskeletal homeostasis maintained by Rho proteins, inducing inflammation and cell death. In contrast, *C. butyricum* inhibits CDI via the following mechanisms: Ⅰ: direct attack by the production of antimicrobial substances, Ⅱ: growth with indigenous bacteria to inhibit the growth of *C. difficile* from nutritional conditions, Ⅲ: inhibition of toxin activity by butyrate, Ⅳ: induction of neutrophils, Th1 and Th17 cells by butyric acid to eliminate *C. difficile*, Ⅴ: activation of IL-17A-producing cells to induce B cells, and the production of IgA to eliminate *C. difficile.* Solid lines indicate mechanisms that have been already reported. Dashed lines indicate expected mechanisms. Arrows indicate active pathways. The T-shaped lines indicate the inhibitory pathway.

**Figure 2 biomedicines-10-00483-f002:**
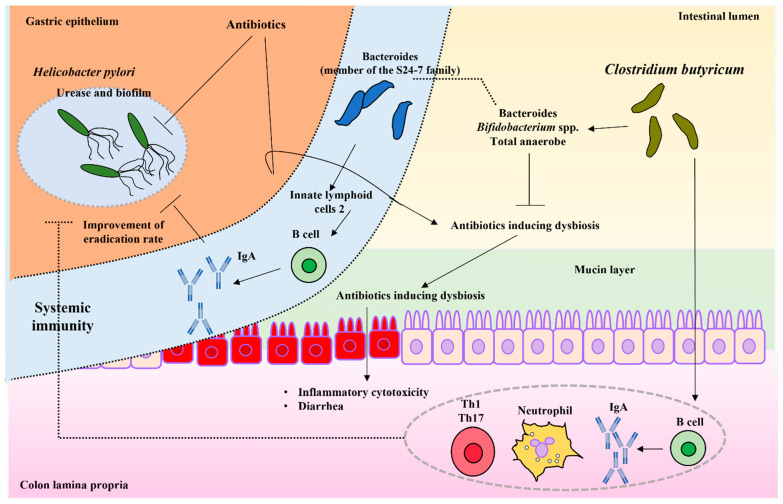
The protective mechanisms from *H. pylori* infection with *C. butyricum*: antibiotic eradication therapy is employed for *H. pylori* infection. As a result, dysbiosis is induced; however, the anaerobic bacteria, *Bacteroides* and *Bifidobacterium* spp. are retained by the administration of *C. butyricum* with no reduction in their numbers. *C. butyricum* increases the eradication rate of *H. pylori* owing to the following reasons: Ⅰ: systemic immunity is activated, which helps to eliminate *H. pylori* in the stomach, Ⅱ: Bacteroides (a member of the S24-7 family) is involved in innate 2 (ILC2) in the stomach, and IL-5 released from ILC2 stimulates B cells to produce IgA. Solid lines indicate mechanisms that have been already reported. Dashed lines indicate expected mechanisms. Arrows indicate active pathways. The T-shaped lines indicate the inhibitory pathway.

**Figure 3 biomedicines-10-00483-f003:**
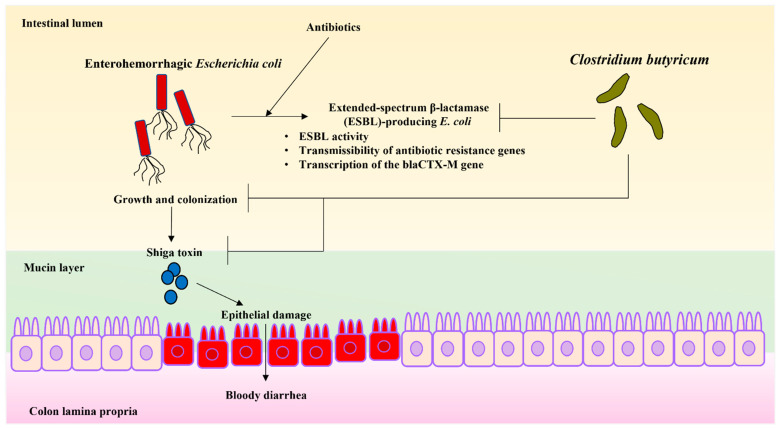
The mechanisms whereby *C. butyricum* protects against enterohemorrhagic *E. coli* (EHEC) infection. EHEC causes hemorrhagic diarrhea. CBM 588 alleviates this symptom by inactivating EHEC growth and toxin. Although infections caused by *E. coli* are treated with antibiotics, the development of resistance is concerning. CBM 588 has been reported to have the following effects on the development of resistance in *E. coli*. Ⅰ: inactivation of ESBLs, Ⅱ: repression of the transcription of the blaCTX-M gene during the growth phase of *E. coli*, Ⅲ: inhibition of the transmissibility of antibiotic resistance genes by enteric bacteria. Solid lines indicate mechanisms that have been already reported. Arrows indicate active pathways. The T-shaped lines indicate the inhibitory pathway.

## Data Availability

Not applicable.

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
