# Peer review of "Effect of Clostridium butyricum on Gastrointestinal Infections"

_biomedicines, 2022, doi:10.3390/biomedicines10020483_

Round 1

Reviewer 1 Report

The authors studies the ability of  C. butyricum strains to inhibit many intestinal pathogenic bacteria growths and to enhance host intestinal immunity but they do not consider its possible antiviral action. It would be appropriate to mention this aspect as well.                                               Another point should be cited is the consideration that gastrointestinal infections should be reduced in individuals colonized by C.butyricum, can the authors address  this aspect?                                                                                                                                    Additional feature to be evaluated is  whether it is appropriate  in case of bacterial diarrhea  to carry out C.butyricum research before therapeutic administration.

Author Response

Revisions required for publication (seen by authors, reviewers, and editors)

The changes for each page are as follows i.e.(Page xx, Line xx).

 To Reviewer #1:

Thank you for your comments and suggestions. We conducted additional works and revised our manuscript in some parts based on each reviewer’s suggestion with blue characters and yellow highlight. Thanks to reviewers’ comments, we think additional experimental data and some revises according to your suggestions enhanced the quality of our manuscript.

Question 1: The authors studies the ability of C. butyricum strains to inhibit many intestinal pathogenic bacteria growths and to enhance host intestinal immunity but they do not consider its possible antiviral action. It would be appropriate to mention this aspect as well.

 Answer: Thank you for your suggestion. The evidence to show the anti-viral effect of C. butyricum has not been confirmed. However, we are expecting that the immunostimulation by CBM 588 may indirectly inhibits virus infection [1,2] through Lactobacillus and Bifidobacterium species proliferations [3-5]. Furthermore, one of fatty acids induced by CBM 588 have an anti-influenza effect by inhibition of viral RNA transport to the nuclear envelope [6]. Thereby, we have added text to the Conclusion section and cited the recommended article. (Page 9, Lines 363-371).

“Although this review has specifically discussed bacterial gastrointestinal infections, some studies have shown that probiotics such as Lactobacillus spp. and Bifidobacterium spp. have antiviral effects [117-119]. The evidence to show the anti-viral effect of C. butyricum has not been confirmed. However, CBM 588 induces cytokines and immune cells with antiviral effects, including IFN-γ and/or IgA, in various models. Hence, CBM 588 may indirectly inhibits virus infection through Lactobacillus and Bifidobacterium spp. proliferations [62,68]. Also, protectin D1, the metabolite induced by CBM 588 in antibiotic causing dysbiosis model, inhibits viral RNA transport to the nuclear envelope, thereby suppressing influenza virus replication [120]. “

[Reference]

  1. Hagihara, M.; Ariyoshi, T.; Kuroki, Y.; Eguchi, S.; Higashi, S.; Mori, T.; Nonogaki, T.; Iwasaki, K.; Yamashita, M.; Asai, N.; et al. Clostridium butyricum enhances colonization resistance against Clostridioides difficile by metabolic and immune modulation. Sci Rep. 2021, 11(1), 15007. doi: 10.1038/s41598-021-94572-z. PMID: 34294848; PMCID: PMC8298451.
  2. Hayashi, A.; Nagao-Kitamoto, H.; Kitamoto, S.; Kim, C. H.; Kamada, N. The Butyrate-Producing Bacterium Clostridium butyricumSuppresses Clostridioides difficile Infection via Neutrophil- and Antimicrobial Cytokine-Dependent but GPR43/109a-Independent Mechanisms. J Immunol. 2021, 206(7), 1576-1585. doi: 10.4049/jimmunol.2000353. Epub 2021 Feb 17. PMID: 33597149; PMCID: PMC7980534
  3. Fujii, T.; Jounai, K.; Horie, A.; Takahashi, H.; Suzuki, H.; Ohshio, K.; Fujiwara, D.; Yamamoto, N. Effects of Heat-Killed Lactococcus Lactis Lactis JCM 5805 on Mucosal and Systemic Immune Parameters, and Antiviral Reactions to Influenza Virus in Healthy Adults; a Randomized Controlled Double-Blind Study. Journal of Functional Foods. 2017, 35, 513–521, doi:10.1016/j.jff.2017.06.011.
  4. Tsuji, R.; Yamamoto, N.; Yamada, S.; Fujii, T.; Yamamoto, N.; Kanauchi, O. Induction of Anti-Viral Genes Mediated by Humoral Factors upon Stimulation with Lactococcus Lactis Strain Plasma Results in Repression of Dengue Virus Replication in Vitro. Antiviral Res. 2018, 160, 101–108, doi:10.1016/j.antiviral.2018.10.020.
  5. Iwabuchi, N.; Xiao, J.-Z.; Yaeshima, T.; Iwatsuki, K. Oral Administration of Bifidobacterium Longum Ameliorates Influenza Virus Infection in Mice. Biol Pharm Bull. 2011, 34, 1352–1355, doi:10.1248/bpb.34.1352.
  6. Morita, M.; Kuba, K.; Ichikawa, A.; Nakayama, M.; Katahira, J.; Iwamoto, R.; Watanebe, T.; Sakabe, S.; Daidoji, T.; Nakamura, S.; et al. The Lipid Mediator Protectin D1 Inhibits Influenza Virus Replication and Improves Severe Influenza. 2013, 153, 112–125, doi:10.1016/j.cell.2013.02.027.

Question 2: Another point should be cited is the consideration that gastrointestinal infections should be reduced in individuals colonized by C. butyricum, can the authors address this aspect?

 Answer: Thank you for your suggestion. By now, no clinical evidence has shown that CBM 588 can reduce pathogenic bacteria in gut by itself. However, there are some supportive in vivo and in vitro study data that CBM 588 would be effective to inhibit the proliferations of bacteria can cause gastrointestinal infections by butyrate production, nutritional competition, and production of antibiotic substances such as bacteriocins [1,2,3]. Additionally, we are thinking that the one of the therapeutic purposes for the probiotics treatment is to inhibit the recurrence and colonization of pathogenic bacteria. Therefore, we added some sentences attached below including the case of prophylactic administration of CBM 588 in clinical study and the case of pre-administration in basic research of EHEC and CDI [4,5]. (Page 8, Lines 320-336).

“By now, no clinical evidence has shown that CBM 588 can reduce pathogenic bacteria in gut by itself. However, there are some supportive in vivo and in vitro study data that CBM 588 would be effective to inhibit the proliferations of bacteria can cause gastrointestinal infections by butyrate production, nutritional competition, and production of antibiotic substances such as bacteriocins [62, 67, 68, 79, 87, 88]. Additionally, we are thinking that the one of the therapeutic purposes to use probiotics is to inhibit the recurrence and colonization of pathogenic bacteria. Then, some clinical studies have suggested that C. butyricum strains not only inhibit many pathogenic bacteria growths, but also enhance host intestinal immunity. In the case of clinical study, the co-administration of CBM 588 with antibiotics before the onset of diarrhea significantly reduced the incidence of diarrhea [27]. Furthermore, in EHEC-infected rabbit models, prior administration of C. butyricum inhibited the growth of EHEC and reduced the incidence of diarrhea and the lethality [104]. In addition, CBM 588 administration prior to CDI induced Th1, Th17, and neutrophils ​and enhanced C. difficile colonization resistance, thereby reducing the lethality [69]. Hence, prior colonization of C. butyricum in gut is expected to inhibit the invasion of pathogenic bacteria and to maintain the homeostasis of the microbiota.”

[Reference]

  1. Yajima, T. In: "Physiological and clinical aspects of short-chain fatty acids" Cummings, J.H.; Rombeau, J.L.; Sakata, T eds., Cambridge University Press, Cambridge. 1995, pp. 209-221.
  2. Nakanishi, S.; Tanaka, M. Sequence analysis of a bacteriocinogenic plasmid of Clostridium butyricum and expression of the bacteriocin gene in Escherichia coli. Anaerobe. 2010, 16(3), 253-7. doi: 10.1016/j.anaerobe.2009.10.002. Epub 2009 Oct 17. PMID: 19840859.
  3. Evans, C. T.; Safdar, N. Current Trends in the Epidemiology and Outcomes of Clostridium difficile Clin Infect Dis. 2015, 60 Suppl 2, S66-71. doi: 10.1093/cid/civ140. PMID: 25922403.
  4. Baj, J.; Forma, A.; Sitarz, M.; Portincasa, P.; Garruti, G.; Krasowska, D.; Maciejewski, R. Helicobacter pyloriVirulence Factors-Mechanisms of Bacterial Pathogenicity in the Gastric Microenvironment. 2020, 10(1), 27. doi: 10.3390/cells10010027. PMID: 33375694; PMCID: PMC7824444.
  5. Tachikawa, T.; Seo, G.; Nakazawa, M.; Sueyoshi, M.; Ohishi, T.; Joh, K. Estimation of probiotics by infection model of infant rabbit with enterohemorrhagic Escherichia coli O157:H7. Kansenshogaku Zasshi. 1998, Dec;72(12):1300-5. Japanese. doi: 10.11150/kansenshogakuzasshi1970.72.1300. PMID: 9916417.

Question 3: Additional feature to be evaluated is whether it is appropriate in case of bacterial diarrhea to carry out C.butyricum research before therapeutic administration. 

 Answer: Thank you for your suggestion. CBM 588 is a very safe bacteria, that is why the bacteria has not toxin producing genes and has been used clinically for over 70 years [1,2]. Additionally, CBM588 has anti-inflammatory effects, mucosal protective effects and inhibits the growth of pathogenic bacteria [3,4]. Therefore, CBM 588 is expected to be effective not only for diarrhea in dysbiosis caused by antibiotics but also for diarrhea in dysbiosis caused by pathogenic bacteria. We added flowing sentences in discussion. (Page 8, Lines 312-319).

“It is also known that the damage to the intestinal tract can be caused by abnormal growth of pathogenic bacteria and associated toxin production [102,103]. Previous in vivo and in vitro study revealed that CBM 588 would be effective to inhibit the proliferations of bacteria can cause gastrointestinal infections [58-68, 75-84, 87-89, 93-95, 98, 100, 101]. Additionally, CBM 588 is a very safe drug since the bacteria has not toxin producing genes and has been used clinically for over 70 years [25-32]. Therefore, CBM 588 is expected to be effective not only for diarrhea in dysbiosis caused by antibiotics but also for diarrhea in dysbiosis caused by pathogenic bacteria.”

[Reference]

  1. The European Commission. Commission Implementing Decision of 11 December 2014 authorising the placing on the market of Clostridium butyricum (CBM 588) as a novel food ingredient under Regulation (EC) No 258/97 of the European Parliament and of the Council (notified under document C (2014) 9345). 2014, 57, 153.
  2. MIYARISAN PHARMACEUTICAL CO., LTD. Clostridium butyricum MIYAIRI strain [Internet]. Clostridium butyricum MIYAIRI strain [cited 2020 Sep 1];

Available from:  http://www.miyarisan.com/english_index.htm

  1. Hagihara, M.; Kuroki, Y.; Ariyoshi, T.; Higashi, S.; Fukuda, K.; Yamashita, R.; Matsumoto, A.; Mori, T.; Mimura, K.; Yamaguchi, N.; et al. Clostridium butyricum Modulates the Microbiome to Protect Intestinal Barrier Function in Mice with Antibiotic-Induced Dysbiosis. 2020, 23(1), 100772. doi: 10.1016/j.isci.2019.100772. Epub 2019 Dec 13. PMID: 31954979; PMCID: PMC6970176.
  2. Ariyoshi, T.; Hagihara, M.; Eguchi, S.; Fukuda, A.; Iwasaki, K.; Oka, K.; Takahashi, M.; Yamagishi, Y.; Mikamo, H. Clostridium butyricum MIYAIRI 588-Induced Protectin D1 Has an Anti-inflammatory Effect on Antibiotic-Induced Intestinal Disorder. Front Microbiol. 2020, 11, 587725. doi: 10.3389/fmicb.2020.587725. PMID: 33193245; PMCID: PMC7661741.

Reviewer 2 Report

The authors have produced a concise review on the “Effect of Clostridium butyricum on gastrointestinal infections”. The review is generally well organised and well written. I suggest that this review would be acceptable for publication in Biomedicines with some minor corrections to the text, as follows:

Line 21.  Please define “AMR” here Lines 22-23.  Italicise “Clostridium butyricum”, “Clostridioides difficile” and “Helicobacter pylori” Line 67.  Insert “as a” before “animals” Line 132.  Remove the “n” before “was” Lines 132-133.  What is meant by “viable CBM 588 contacted to the C. difficile cells”? Line 213.  Change “Helicobacter pylori” to “H. Pylori” Line 258.  Change “Escherichia coli” to “E. coli” Line 290.  Change “Vibrio cholerae” to “V. cholerae” Line 303.  Define “SPF” Lines 360-656.  Please check that all references have the same format and are consistent with each other, especially: Use/non-use of capital letters in article titles. Use italics for names of bacteria. Full or abbreviated journal titles and use/non-use of punctuation in abbreviated journal titles. Spacing between year(bold), volume(issue), page range. Remove the month and day of publication where given. Full or abbreviated page range, i.e. 2251-2260 or 2251-60, 1121-30 or 1121-1130, etc.  

Author Response

Revisions required for publication (seen by authors, reviewers, and editors)

The changes for each page are as follows i.e.(Page xx, Line xx).

Reviewer #2:

Comments: The authors have produced a concise review on the “Effect of Clostridium butyricum on gastrointestinal infections”. The review is generally well organized and well written. I suggest that this review would be acceptable for publication in Biomedicines with some minor corrections to the text, as follows: …

To Reviewer #2: Thank you for your comments and suggestions. We revised our manuscript in some parts according to reviewer’s suggestions with red characters and yellow highlight. Thanks to reviewers’ comments, we think the quality of our manuscript was enhanced.

Comment: Line 21. Please define “AMR”

Response: Thank you. We define AMR as antimicrobial-resistance (AMR) and revised the part. (Page 1, Line 21)

Comment: Lines 22-23. Italicise “Clostridium butyricum”, “Clostridioides difficile” and “Helicobacter pylori”

Response: Thank you. We revised the words as you pointed out. (Page 1, Line 23,24).

Comment: Line 67. Insert “as a” before “animals”

Response: Thank you. We revised the sentence as you pointed out. (Page 2, Line 68).

Comment: Line 132. Remove the “n” before “was”

Response: Thank you. We removed the word as you pointed out.

Question 1: Line 132. What is meant by “viable CBM 588 contacted to the C. difficile cells”?

Response: Viable CBM 588 means living and growth cells. It is considered that the direct contact with the metabolites of CBM 588 suppressed the toxin production and growth of C. difficile. We revised the parts as follow (Page 3, Line 133-136).

“Interestingly, these effects showed only at viable CBM 588, which mean living and growth cells, contacted to the C. difficile cells. And, it is considered that the direct contact with the metabolites of CBM 588 suppressed the toxin production and growth of C. difficile. Such findings suggest that CBM 588 attenuated symptoms related to CDI (Figure 1).”

Comment: Line 213. Change “Helicobacter pylori” to “H. Pylori”

Response: Thank you. We revised the word as you pointed out. (Page 6, Line 215).

Comment: Line 258. Change “Escherichia coli” to “E. coli” (Page 7, Line 258).

Response: Thank you. We revised the word as you pointed out.

Line 290. Change “Vibrio cholerae” to “V. cholerae” (Page 7, Line 290).

Response: Thank you. We revised the word as you pointed out.

Line 303.  Define “SPF”

Response: Thank you. We define SPF as specific pathogen free (SPF) and revised the part. (Page 8, Line 303).

Comments: Lines 360-656.  Please check that all references have the same format and are consistent with each other, especially: Use/non-use of capital letters in article titles. Use italics for names of bacteria. Full or abbreviated journal titles and use/non-use of punctuation in abbreviated journal titles. Spacing between year(bold), volume(issue), page range. Remove the month and day of publication where given. Full or abbreviated page range, i.e. 2251-2260 or 2251-60, 1121-30 or 1121-1130, etc.

Response: We carefully revised the references to be shown with a consisted format, especially remove the month and day of publication in conference poster presentation. (Page 11, Line 478,481,489). (Page 14, Line 652).